# Design of Nanostructured Hybrid Electrodes Based on a Liquid Crystalline Zn(II) Coordination Complex-Carbon Nanotubes Composition for the Specific Electrochemical Sensing of Uric Acid

**DOI:** 10.3390/nano12234215

**Published:** 2022-11-27

**Authors:** Sorina Negrea, Adelina A. Andelescu, Sorina Ilies (b. Motoc), Carmen Cretu, Liliana Cseh, Mircea Rastei, Bertrand Donnio, Elisabeta I. Szerb, Florica Manea

**Affiliations:** 1Department of Applied Chemistry and Engineering of Inorganic Compounds and Environment, Politehnica University of Timisoara, Bvd. Vasile Parvan No. 6, 300223 Timisoara, Romania; 2National Institute of Research and Development for Industrial Ecology (INCD ECOIND), Timisoara Branch, 300431 Timisoara, Romania; 3“Coriolan Drăgulescu” Institute of Chemistry, Romanian Academy, 24 Mihai Viteazu Bvd., 300223 Timisoara, Romania; 4Institut de Physique et Chimie des Matériaux de Strasbourg (IPCMS), CNRS-Université de Strasbourg (UMR7504), 67034 Strasbourg, France

**Keywords:** Zn(II), metallomesogen, nanocomposite paste electrode, carbon nanotubes, sensor, uric acid, cyclic voltammetry

## Abstract

A metallomesogen based on an Zn(II) coordination complex was employed as precursor to obtain a complex matrix nanoplatform for the fabrication of a high-performance electrochemical hybrid sensor. Three representative paste electrodes, which differ by the weight ratio between Zn(II) metallomesogen and carbon nanotubes (CNT), i.e., PE_01, PE_02 and PE_03, were obtained by mixing the materials in different amounts. The composition with the largest amount of CNT with respect to Zn complex, i.e., PE_03, gives the best electrochemical signal for uric acid detection by cyclic voltammetry in an alkaline medium. The amphiphilic structure of the Zn(II) coordination complex likely induces a regular separation between the metal centers favoring the redox system through their reduction, followed by stripping, and is characterized by enhanced electrocatalytic activity towards uric acid oxidation. The comparative detection of uric acid between the PE_03 paste electrode and the commercial zinc electrode demonstrated the superiority of the former, and its great potential for the development of advanced electrochemical detection of uric acid. Advanced electrochemical techniques, such as differential-pulsed voltammetry (DPV) and square-wave voltammetry (SWV), allowed for the highly sensitive detection of uric acid in aqueous alkaline solutions. In addition, a good and fast amperometric signal for uric acid detection was achieved by multiple-pulsed amperometry, which was validated by urine analysis.

## 1. Introduction

Materials based on Zn for electrochemical sensing of uric acid were mainly reported as ZnO nanoparticles (NPs) [1,2], nanoalloys [3,4], spinel-type oxides (Zn_2_Co_2_O_4_) [5] and metal organic frameworks (MOFs) [6]. In order to achieve superior electrochemical performances, various strategies have been employed: NPs size, shape and/or morphology control(s), introduction of a second metal in MOFs [7], hierarchical structuration or composite formation [8,9,10,11,12,13], etc. When the composite is formed with carbon-based materials, the enhancement of the electrical conductivity is found to contribute to the substantial enhancement of the electrochemical sensing parameters [14,15,16,17] in terms of the detection potential value by avoiding ohmic drop. Moreover, nanostructured carbon nanotubes (CNTs) as matrix for metal-based composition of the sensor can act as a substrate, to preconcentrate the analyte to the electrode surface. This improves the limit of detection or response, and acts as an electrocatalyst that enhances the kinetics of the electron transfer at the electrode-analyte interface, with a positive effect on the sensitivity [18,19].

The fabrication of CNT-based electrodes for the electrochemical sensing of analytes are carried out mostly by coating the electrodes by drop casting techniques of dispersions containing CNTs [20]. Carbon paste electrodes avoid the drawbacks of solvent use [21], and are highly advantageous due to their chemical inertness and robustness, renewability, stable response, low ohmic resistance, environmentally friendly and non-toxic characters [22]. Modified carbon paste electrodes are typically a concoction of a non-electrolytic binder, powdered graphite and a non-innocent modifier. These are obtained by alternative methods such as mechanical amalgamation, soaking graphite particles into a solution of modifier, or mixed methods [22,23,24,25,26].

As shown by Cu(I) and Zn(II) coordination complexes [27,28,29], metallomesogens may be excellent sources with which to generate materials with separate metal oxide centers, hence with increased electrochemical performances. Metallomesogens are amphiphilic molecules, having a central metal ion surrounded by organic ligands [30,31,32]. The grafting of several long alkyl chains onto the ligands, needed to induce the required fluidity for obtaining supramolecular-ordered liquid crystalline architectures, implies a separation of the metal centers at the nanometric scale. Moreover, their positions and orderings may be controlled by the mesophase type and symmetry through molecular engineering.

Here, we employed a recently reported metallomesogen based on a pentacoordinated Zn(II) metal center (Zn_MM in Figure 1) that are arranged into smectic phases [29], aiming at obtaining a sensor with advanced electroanalytical parameters for the selective detection of uric acid (UA).

Carbon-based electrochemical sensors are commonly used for non-enzymatic detection of UA because of their fast analysis and simple experimental procedures. However, a modification is required to avoid the interferences with dopamine (DA), glucose (GL) and ascorbic acid (AA), which are present in biological fluids, e.g., blood and urine [33,34]. UA can be easily oxidized into allantoin as the main oxidation product at higher or lower potential values, depending on the electrocatalytic activity of the electrode and the oxidation potentials of DA, AA and GL that can be too close to be separated. Therefore, in order to obtain an accurate determination of UA in the presence of DA, GL and AA, it is compulsory to develop a selective and facile method for a routine assay. In general, the electrode composition is the main factor for achieving selective electrochemical detection related to the role electrocatalyst, only for the UA oxidation process or other types of electrochemical processes (e.g., complex formation) generating the electrochemical response [29]. In this work, we present the design of an optimized paste electrode composition consisting of Zn_MM mixed in a carbon nanotubes (CNTs) matrix through a paraffinic oil for the paste consistency, in order to develop a simple selective amperometric detection of UA in the presence of DA, GL and AA. To the best of our knowledge, no selective determination of UA at negative potential values as low as (−1.20 and −1.50 V/SCE) via anodic and respective cathodic response has yet been reported [2]. For the optimization of the paste electrode content, three different Zn_MM:CNT ratios were actually considered. In order to relate the structure of the final hybrid material to electrochemical properties, the mesomorphic and structural properties of these Zn/CNT/oil mixtures were determined by differential scanning calorimetry (DSC), small and wide-angle X-ray scattering studies (SAXS/WAXS), and compared with those exhibited by the pristine Zn_MM.

## 2. Materials and Methods

### 2.1. Materials

Zn_MM was obtained as previously reported [29]. Multiwall carbon nanotubes (CNT) synthesized by catalytic carbon vapor deposition (CCVD) were purchased from NanocylTM, Sambreville, Belgium.

### 2.2. Synthesis of Hybrids

The modified paste electrodes were obtained as follows: To a solution of 500 mg Zn_MM was dissolved in 7 mL of hexane, and the appropriate quantities of CNT and paraffinic oil were added (see Table 1). The resulting black mixtures were sonicated for 30 min. Then, the solvent was left to evaporate at room temperature, yielding black tar-like waxy solids. The materials were further mechanically mixed in a pestle mortar and dried under vacuum.

### 2.3. Electrochemical Tests

The conventional three-electrodes system consisted of PE_0i, commercial Zn and CNT paste as working electrodes, a platinum plate with the geometrical surface of 1 cm^2^ as counter-electrode, and a saturated calomel (SCE) reference electrode, coupled at Autolab potentiostat/galvanostat PGSTAT 302 (Eco Chemie, Utrecht, The Netherlands) controlled with Nova 2.2 software, was used for all electrochemical measurements. The cyclic voltammetry (CV) technique was used for the electrochemical characterizations of paste electrodes considering electroactive surface areas and for the electrochemical behavior of uric acid (UA). Two levels-based chronoamperometry (CA) and two/four-pulsed amperometry techniques were employed for the selective detection of uric acid in the presence of dopamine (DA), glucose (GL) and ascorbic acid (AA). Uric acid, DA, GL and AA were used as received from Aldrich, Darmstadt, Germany. 0.1 M NaOH supporting electrolyte was prepared by using NaOH and distilled water. For each electrochemical determination, the working electrode was stabilized by three repetitive cyclic voltammetry scanning, within the potential range from −1.50 to +1.00 V/SCE at the scan rate of 0.05 V∙s^−1^.

### 2.4. Characterisation Techniques

Transition temperatures and enthalpies were recorded with a Q1000 apparatus from TA Instruments, Paris, France. The instrument was calibrated with indium and otherwise stated three heating/cooling cycles were performed on each sample, with a heating and cooling rate of 10 °C/min. SAXS/WAXS patterns were obtained with a transmission Guinier-like geometry. A linear focalized monochromatic Cu Kα1 beam (λ = 1.5405 Å) was obtained using a sealed-tube generator (600 W) equipped with a bent quartz monochromator. In all cases, samples were filled in home-made sealed cells of 1 mm path. The patterns were recorded with a curved Inel CPS120 counter gas-filled detector, Paris, France, linked to a data acquisition computer (periodicities up to 90 Å) and on image plates scanned by Amersham Typhoon IP, Cytiva, France with 25 μm resolution (periodicities up to 120 Å).

## 3. Results

Three different content paste electrodes were obtained, labeled PE_01, PE_02 and PE_03, with a different weight ratio of Zn(II) metallomesogen and CNT. The compositions are shown in Table 1. The paste electrodes were obtained by sonicating a solution of the metallomesogen, CNT and paraffinic oil in hexane for 30 min. After evaporation of the solvent, the materials were further mechanically mixed in a pestle mortar and dried under vacuum. The preparation process of CNT-based hybrids and paste electrodes is described in Figure 2.

### 3.1. Mesomorphic Properties

#### 3.1.1. DSC Studies

The thermal behavior of the paste electrodes PE_0i was investigated by DSC, and the transition temperatures and enthalpies of transitions compared to those of the pristine Zn_MM previously reported [29]. Zn_MM was obtained as a crystalline solid, with the transition to a SmA mesophase at 94.3 °C and a clearing point at 116.5 °C (see Table 2 and Figure 3). On cooling from the isotropic liquid, the molecules first self-assemble into a SmA mesophase, followed by crystallization, retaining some structural features of the smectic phase (therefore labeled as CrSm).

The DSC traces and transition temperatures measured on the paste electrodes thermograms were not expected to be very different to those derived from the pristine metallomesogen, since neither the paraffinic oil nor CNT show transitions in this temperature region. Therefore, the associated enthalpies were calculated using the molecular weight of Zn_MM.

Indeed, the thermal behavior of the paste electrodes resembles that of the pristine Zn_MM with the loss of the crystalline phase of the pure Zn complex (at around 60 °C). The SmA still occurs in the paste electrodes upon heating, but with a broader phase transition and a slight decrease of the transition temperatures. This is in agreement with the increasing amount of both CNT and oil at the detriment of the amount of complex. The isotropization temperature is not detected by DSC, however. On cooling, the isotropic liquid to SmA phase transition is still not detected, and the transition to the crystalline state is only clearly seen for the PE_01 mixture. The thermal behavior of the three mixtures is similar in the second heat/cool cycle. Thus, it can be concluded that the mesophase is still present in the paste electrodes but, as the pure complex, it is crystallized at room temperature in a smectic phase.

#### 3.1.2. S/WAXS Studies

The confirmation of the organization of paste electrodes was achieved by small- and wide-angle X-ray scattering studies recorded at room temperature (SAXS/WAXS—Figure 4). The patterns of the three mixtures are similar with the one collected for the pristine Zn_MM, with no contribution detected from CNT. For all samples, at room temperature the patterns agreed with the formation of crystalline phase retaining the layering of the smectic phase, CrSm phase (Figure 3). In the wide-angle range, chain and complex crystallization was confirmed for all samples.

### 3.2. Electrochemical Studies

#### 3.2.1. Electrochemical Characterization of Paste Electrodes (PE_0i)

Before UA electrooxidation, the electroactive surface areas were determined for each composition using a classical ferri/ferrocyanide system, described in detail in our previous publications [35,36,37]. A higher content of carbon nanotubes and lower content of Zn_MM required a higher content of paraffinic oil to ensure a mechanical stability of the electrode and better distribution of CNT. This also slightly enhanced the electroactive surface area (see Table 3).

#### 3.2.2. Electrochemical Behavior of Uric Acid (UA) on Paste Electrodes (PE_0i)-Cyclic Voltammetry Study

The results of the cyclic voltammetrical behavior for the detection of UA at different concentrations (1 to 5 mM) on paste electrodes, pure CNT paste electrode and commercial Zn electrode by CV within the potential ranged from −1.50 to +1.00 V vs. SCE are shown and compared in Figure 5a–e. A very interesting electrochemical behavior is found for the PE_0i electrodes at the cathodic branch of the CV (−1.50 to −1.2 V/SCE) corresponding to the redox system of Zn and ZnO formation and its stripping [29]. This is affected by the presence of UA through formation of Zn:uric acid complexes and its decomposition [38]. This behavior is found only for the PE_0i electrodes and is not observed for the Zn metallic electrode (Figure 5d). Most likely, the difference resides in the Zn mobility in the PE_0i electrodes and the background currents corresponding to the capacitive component, which is lower in comparison with that generated by CNT. The anodic signal recorded at −1.25 V/SCE increases linearly with UA concentration, and the best sensitivity is reached for PE_03 (Figure 6a). The cathodic signal recorded at −1.50 V/SCE increases linearly with UA concentration, increasing only for PE_02 and PE_03, while for PE_01 no linear dependence is found (Figure 6a). This behavior confirms the importance of the zinc:urate complex in the Zn_MM:CNT mixed paste electrode composition, revealed by the manifestation of the synergy effect. CNT acts as a support for the redox system of Zn and ZnO formation and its stripping within the cathodic branch, allowing both anodic and cathodic signals for UA detection. The clear oxidation process of UA is manifested within the anodic branch for all electrodes at different potential values and sensitivities (see Figure 6a–d).

As stated above, a very important aspect in voltammetric and amperometric sensing is represented by selectivity to avoid potential interferences of other biological molecules. The electrochemical behaviors of DA, GL and AA were tested in comparison with UA for all compositions of PE_0i (see Appendix A). DA exhibited significant interferences for PE_01 (DA:UA sensitivity ratio is 5.44), which decreased for PE_02 (DA:UA sensitivity ratio is 1.29), and the lowest was manifested for PE_03 (DA:UA sensitivity ratio is 0.97). No interferences of GL and AA were found for PE_0i electrodes. All of the above results reclaim PE_03 for further developing an optimized detection method.

To characterize more in depth, the electrochemical behavior of UA onto PE_03 electrode, the scan rate influence, ranging from 0.01 to 0.3 V·s^−1^, was also studied considering the electrochemical behavior of the PE_03 electrode in 0.1 M NaOH supporting electrolyte. The complete results are presented in Appendix A, and Figure 7a,b which present CVs recorded at the lowest scan rate of 0.01 V·s^−1^ and the highest scan rate of 0.3 V·s^−1^. According to our previously reported data related to the electrochemical behavior of Zn_MM [29], UA presence affects the redox system of Zn and ZnO formation and its stripping and respective dissolution and also, the inner and outer dissolved O_2_ reduction. This is more visible at the low versus the high scan rate due to the low kinetics of the processes. However, the redox system of Zn and ZnO formation and its stripping is more visible, which confirms the fast kinetics in comparison with other redox systems. All processes are diffusion-controlled in both the presence and absence of UA (based on linear dependence between the currents recorded at each potential value vs. the square root of the scan rate, Appendix A), which make them good candidates for detection. No significant shifting for the detection potentials were found in the presence of UA (see Appendix A).

In order to explore the importance of the structural characteristics of the PE_03 electrode, CV spectra in the presence of various concentrations of UA were recorded at 80 °C, where the Zn-MM component is in the SmA phase. No redox system of Zn and ZnO formation and its striping were found (See Appendix A). After cooling the electrode, the electrode surface structure and its electrochemical behavior returned to the previous state after 48 h, which shows the electrode stability and reproducibility and the importance of having an organized but not fluid structure.

#### 3.2.3. Development of Selective Amperometric Detection of UA

It is well-known that amperometric detection is the most useful for practical applications because of its ease of operation. The chronoamperograms recorded at the potential value of −1.50 and −1.25 V/SCE are presented in Figure 8. These potential values were selected considering the redox system of Zn and ZnO formation and its stripping. The steady-state cathodic responses increase linearly with UA concentration (inset of Figure 8), while no linear dependence was found for steady-state anodic responses (not shown). In addition, a very low sensitivity is obtained with respect to the one obtained by CV (0.8 vs. 26.29 µA∙mM^−1^).

To improve the sensitivity, a multiple-pulsed amperometry (MPA) technique was considered, starting with the two potential levels tested above by CA. The two-pulsed amperograms recorded for PE_03 are presented in Figure 9a. A different time pulse is used to favor the anodic response recorded at −1.20 V/SCE. Thus, the first potential pulse value of −1.50 V/SCE (duration of 0.1 s) at which Zn and ZnO formation occurs, and the second potential pulse value of −1.20 V/SCE (duration 0.05 s), correspond to the stripping process and Zn-urate complex formation. Both allow the achievement of a better sensitivity for the cathodic response compared to the one recorded by CA. In particular, the linear anodic response with increasing UA concentration was characterized by more than twice the sensitivity obtained for the cathodic side (Figure 9b).

Due to the possible interference of DA for UA detection, a two-pulsed amperometry technique was used. Different MPA schemes were optimized by operating under various conditions related to the pulse number and corresponding pulse duration. The optimized four-pulsed potential levels scheme consisted of:
−1.50 V/SCE (pulse time of 0.1 s) for electrode surface renewing by Zn and ZnO formation−1.20 V/SCE (pulse time of 0.03 s) where Zn^2+^ is stripping and Zn:urate complex is forming+0.10 V/SCE (pulse time of 0.05 s) where UA oxidation is starting+0.70 V/SCE (pulse time of 0.05 s) corresponding to the zinc oxide dissolution with the zincate formation, which is controlled by the HO^-^ anions diffusion at the electrode surface [29] and UA oxidation.

The four-pulsed amperograms recorded at the PE_03 paste electrode are presented in Figure 10a. At the potential levels of +0.10 and +0.70 V/SCE, at which UA oxidation process occurs, the DA (at 1 mM) interfered with the UA detection, while for the cathodic branch (−1.50 and respective, −1.20 V/SCE) no interference is noticed. It is worth mentioning that only at the pulse time of 0.03 s for E = −1.20 V/SCE was the interference avoided. This aspect might be explained by the different kinetics of the Zn and ZnO formation and its stripping, in agreement with the results of the scan rate influence using CV.

The optimized four-pulsed amperometry allowed to obtain the best sensitivities (Figure 10b), and the detection of the lowest amounts of UA of 25 µM at the potential value of −1.20 V/SCE and of 33 µM at the potential value of −1.50 V/SCE.

In comparison with some reported electrochemical UA sensors, which are based on UA oxidation presented in Table 4, the PE_03 sensor exhibits higher sensitivity using amperometry; that is the most practical technique and negative detection potential at which Zn:urate complex is formed, assuring its selectivity. The LOD of 25 µM for UA detection is appropriate for biological application.

The good reproducibility of PE_03 for detecting UA was determined by repetitive measurements, with three electrodes in 0.1 M NaOH solution containing 1 mM UA using an optimized four-pulsed amperogram: the RSD for UA detection was 2%. The stability of the PE_03 was tested on the 1st, 7th, 15th and 30th day in a 0.1 M NaOH solution containing 1 mM UA and about 96.2% of the initial current was retained, which proved the long-term stability of the electrode.

The application of the PE_03 paste electrode using four-pulsed amperometry to determine the UA content in real human urine samples was also studied. The urine sample was mixed with 1 M NaOH solution to obtain 0.1 M NaOH in real urine. The optimized four-pulsed amperometry was applied in the mixture of 0.1 M NaOH with real urine as the supporting electrolyte, and known concentrations of uric acid were added. All responses increased as the UA concentration was increased. In the same supporting electrolyte, three successive additions of 10 mL real urine also led to an increase for all responses and, based on the calibration for UA, the content of uric acid in real urine was determined and validated by the results of the medical laboratory (400 mg UA/dm^3^).

## 4. Conclusions

In the present work, the potential of using an amphiphilic Zn(II) coordination complex for obtaining performant non-enzymatic electrochemical sensor was explored. Here, Zn_MM was distributed within CNTs and paraffin oil in three compositions to fabricate paste electrodes. Within the electrodes, the complexes are arranged into a smectic crystalline structure, and the regular separation of the metal centers likely is the key feature in improving the performances of the electrodes. Their organization at room temperature was determined in relation to the pristine metallomesogen.

These paste electrodes were developed and applied for the selective detection of UA. Based on the electroactive surface area and the best signal obtained, an optimum composition was determined, which is Zn_MM:CNT:parrafin oil = 6.25:1:3, named PE_03 paste electrode. The unique properties of Zn_MM and CNT in terms of redox characteristics in the cathodic branch (−1.50 and −1.20 V/SCE) allowed for obtaining both cathodic and anodic responses, probably through formation of Zn:uric acid complexes and their decomposition. Besides the cathodic branch, UA oxidation was found in the anodic branch, which is common with other reported sensors for UA detection. The four-pulsed amperometry technique, optimized in relation to the potential and the number of pulses, was able to selectively detect UA at the potential values of −1.50 through cathodic signal and respective, −1.20 V/SCE by anodic signal. This result is remarkable since the interferences of DA, GL and AA were largely avoided. PE_03 paste electrode exhibited excellent results with the LOD of 25 µM at the detection potential value of −1.20 V/SCE and 33 µM at the detection potential value of −1.50 V/SCE, respectively. Hence, PE_03 can act as an excellent sensor for the selective determination of UA in the presence of DA, GL and AA. The PE_03 electrode, in conjunction with the four-pulsed amperometry, can be successfully applied for the detection of UA in real human urine samples.

The performances of the present paste electrode for electrochemical sensing of uric acid, derived from the optimization of the ratio between CNT and Zn_MM and the structural organization of the ZnO, centers in a non-fluid smectic phase. The latter is related to the absence of the electrochemical response in the SmA mesophase and the reproductible response after 48 h cooling. The research towards obtaining performant non-enzymatic electrochemical sensors based on metallomesogens may bring important advancements in the field. This is partly due to the facility of changing organizations of the metal centers at room temperature by controlling the type and symmetry of the mesophases, which template the structuration at room temperature. Finally, the ability of the amphiphilic Zn(II) complex to self-assemble into different symmetry liquid crystalline phases as a function of the functionalities grafted on the *tpy* ligand [44], likely enables further studies of relevance for structure-related electrochemical properties.

## Figures and Tables

**Figure 1 nanomaterials-12-04215-f001:**
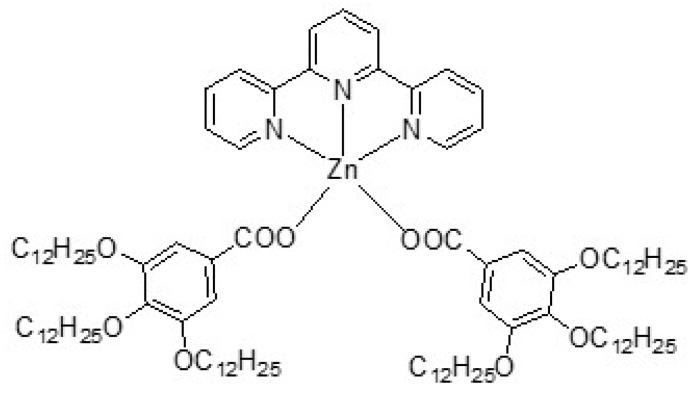
Chemical structure of the complex Zn_MM.

**Figure 2 nanomaterials-12-04215-f002:**
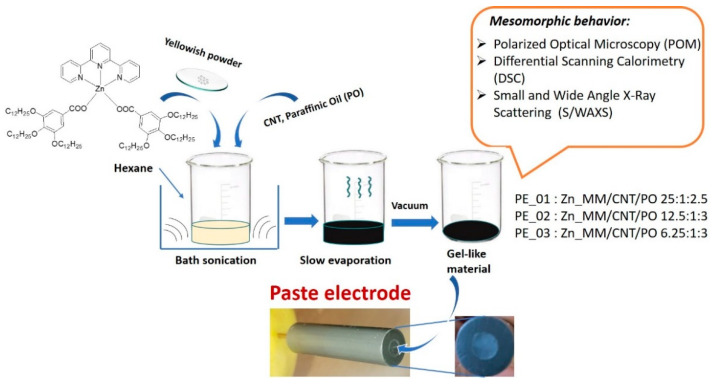
Schematic describing the preparation process of the hybrids and paste electrodes.

**Figure 3 nanomaterials-12-04215-f003:**
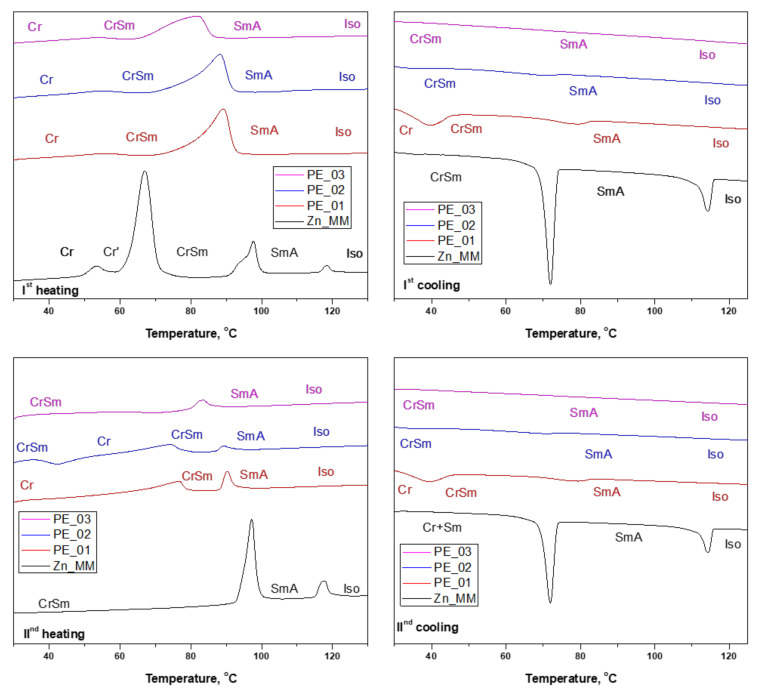
DSC thermograms of pristine complex Zn_MM and paste electrodes PE_0i (Top and bottom rows: 1st and 2nd heat/cool cycle, respectively).

**Figure 4 nanomaterials-12-04215-f004:**
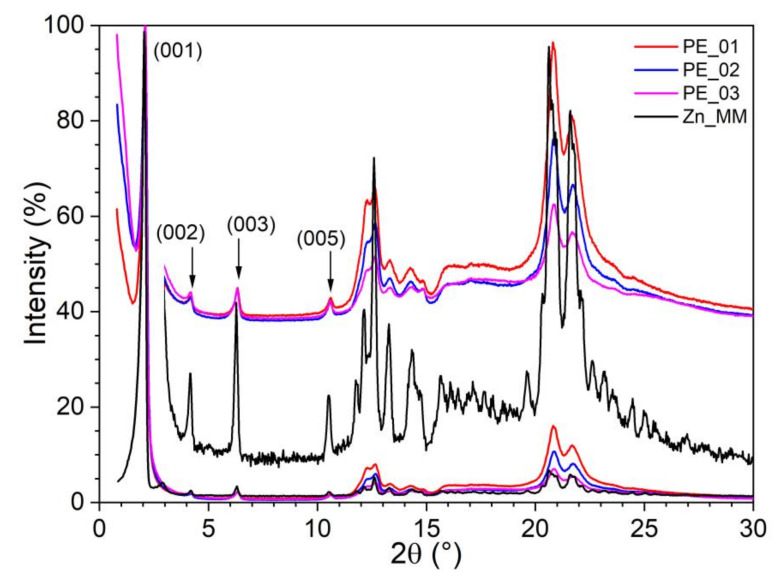
SAXS/WAXS patterns of pristine complex Zn_MM and paste electrodes PE_0i.

**Figure 5 nanomaterials-12-04215-f005:**
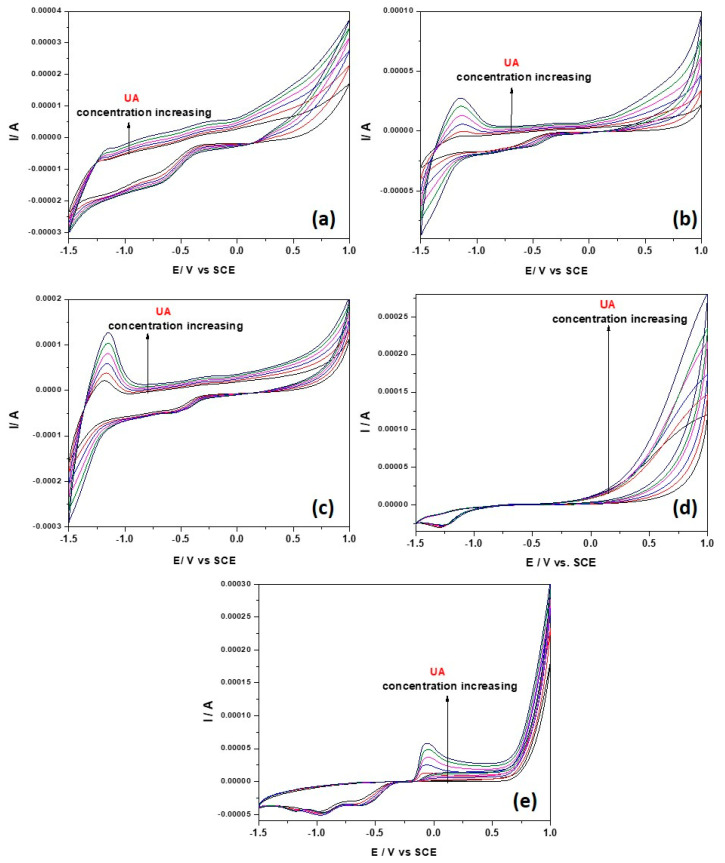
Cyclic voltammograms recorded with potential scan rate of 0.05 V·s^−1^ within potential range from −1.50 V to +1.00 V/SCE in 0.1 M NaOH supporting electrolyte (black line) and in the presence of 1–5 mM UA concentrations at the electrodes (colored lines): (**a**) PE_01, (**b**) PE_02, (**c**) PE_03, (**d**) commercial Zn electrode and (**e**) CNT paste electrode.

**Figure 6 nanomaterials-12-04215-f006:**
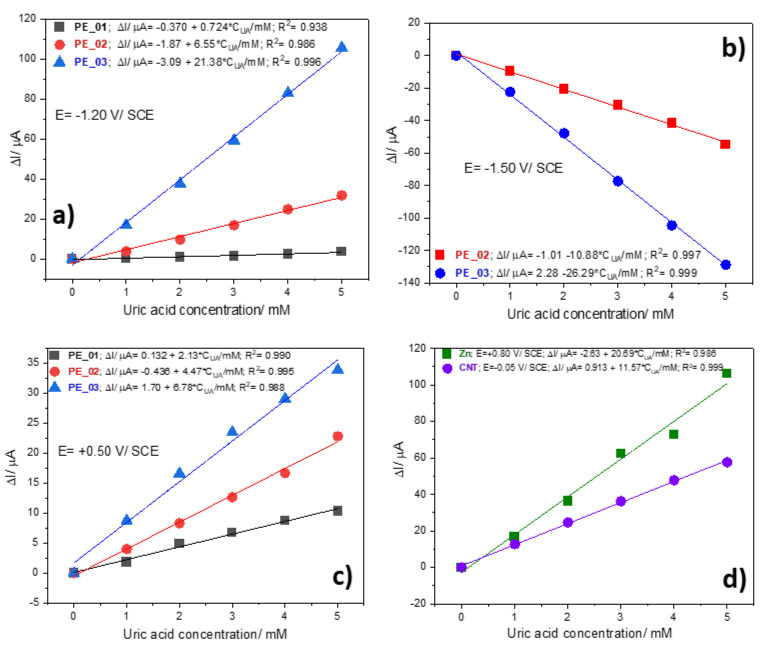
Calibration plots of useful current vs. UA concentrations recorded at: (**a**) E = −1.20 V/SCE on PE_01, PE_02, and PE_03 electrodes; (**b**) E = −1.50 V/SCE on PE_02 and PE_03 electrodes; (**c**) E = +0.50 V/SCE on PE_01, PE_02, and PE_03 electrodes; (**d**) E = +0.80 V/SCE on commercial Zn electrode, and E = −0.05 V/SCE on CNT paste electrode.

**Figure 7 nanomaterials-12-04215-f007:**
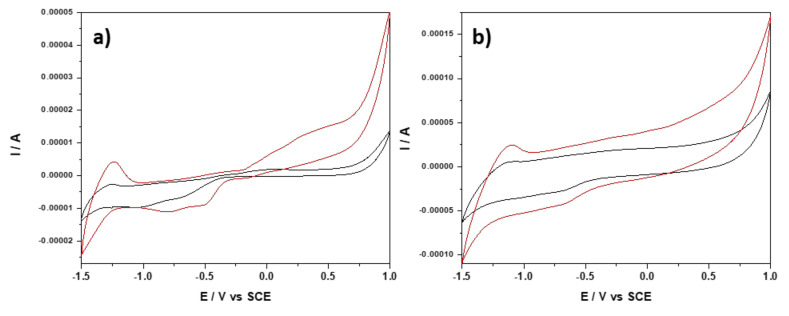
CVs recorded with PE_03 within the potential range of −1.50 to +1.00 V/SCE in 0.1 M NaOH supporting electrolyte (black curve) and in the presence of 2 mM UA (red curve) at the scan rate: (**a**) 0.01 V·s^−1^ and (**b**) 0.3 V·s^−1^.

**Figure 8 nanomaterials-12-04215-f008:**
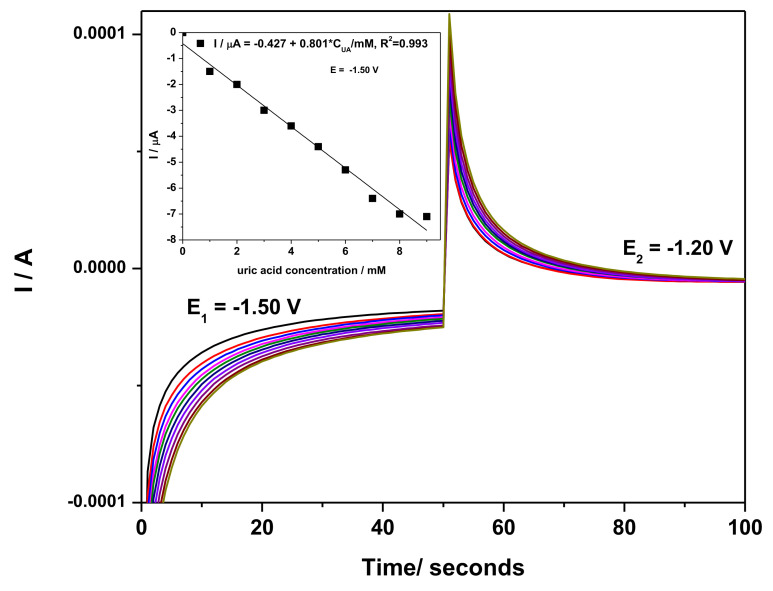
Two detection potential levels-based chronoamperometry recorded at PE_03 paste electrode in 0.1 M NaOH supporting electrolyte (black line) and in the presence of 1–9 mM UA concentrations at the electrodes (colored lines).; Inset: Calibration plots of currents vs. AU concentrations at E = −1.50 V/SCE; no linear dependence was found at −1.20 V/SCE.

**Figure 9 nanomaterials-12-04215-f009:**
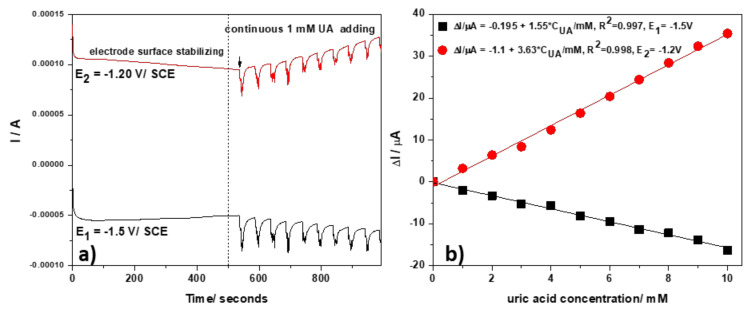
(**a**) Multiple-pulsed amperograms (MPAs) recorded with the PE_03 paste electrode in 0.1 M NaOH supporting electrolyte and in the presence of continuous adding 1 mM UA; the detection potential levels of: E1 = −1.50 V/SCE (pulse time of 0.1 s), and E2 = −1.20 V/SCE (pulse time of 0.05 s). (**b**) Calibration plots recorded at: E1 = −1.50 V/SCE (black) and E2 = −1.20 V/SCE (red).

**Figure 10 nanomaterials-12-04215-f010:**
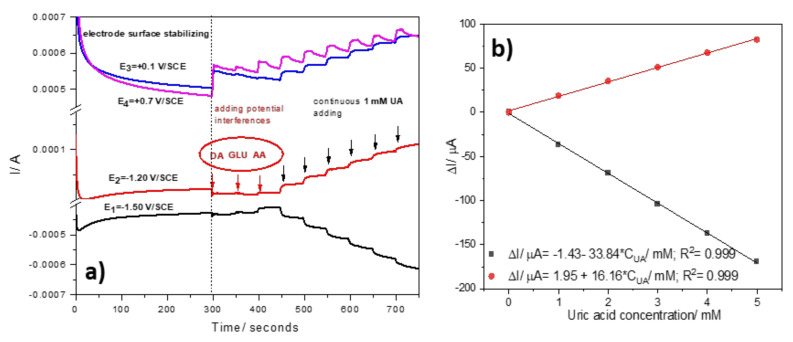
(**a**) Four-pulsed amperograms (MPAs) recorded with the PE_03 paste electrode in 0.1 M NaOH supporting electrolyte in the presence of potential interferences (1 mM) and continuous adding of 1 mM UA; the detection potential levels of: E1 = −1.50 V/SCE, and E2 = −1.20 V/SCE. (**b**) Calibration plots recorded at: E1 = −1.50 V and E2 = −1.20 V/SCE.

**Table 1 nanomaterials-12-04215-t001:** Composition of paste electrodes PE_0i.

Paste Electrode	Zn_MM		CNT		Paraffinic Oil	
	Weight Ratio [%]	Mass [mg]	Weight Ratio [%]	Mass [mg]	Weight Ratio [%]	Mass [mg]
PE_01	25	500	1	20	2.5	50
PE_02	12.5	500	1	40	3	120
PE_03	6.25	500	1	80	3	240

**Table 2 nanomaterials-12-04215-t002:** Transitions temperatures and enthalpy for the pure Zn complex (Zn_MM) and the different mixtures.

Cpd.	Heating/ Cooling Cycle	Mesophases,^1^ Transition Temperatures (°C) and Enthalpies [ΔH (kJ·mol^−1^)]
Zn_MM	I II	Cr 49.0 [3.8] Cr’ 62.5 [53.0] CrSm 94.3 [13.5] SmA 116.5 [1.6] Iso Iso 115.7 [−2.4] SmA 73.7 [−7.4] CrSm CrSm 89 [12.0] SmA 109 [2.5] Iso Iso 114 [−2.6] SmA 73 [−10.2] CrSm
PE_01	I II	Cr 36.2 [5.7] CrSm 80.4 (43.2) SmA [-] ^2^ Iso Iso [-] ^2^ SmA 82.6 [−0.9] CrSm 45.0 [−3.5] Cr Cr 64.3 [−4.4] CrSm 80.4 [2.4] SmA [-] ^2^ Iso Iso [-] ^2^ 82.8 [−0.7] SmA 45.1 [−3.7] CrSm
PE_02	I II	Cr 35.9 [7.2] CrSm 78.5 [45.5] SmA [-] ^2^ Iso Iso [-]^2^ 74.3 [−0.5] SmA [-]^2^ CrSm CrSm 37.2 [−3.9] Cr 61.9 [4.2] CrSm 87.2 [1.0] SmA [-] ^2^ Iso Iso [-]^2^ SmA 74.0 [0.5] CrSm
PE_03	I II	Cr 34.2 [5.9] CrSm 66.2 [43.2] SmA [-] ^2^ Iso Iso [-] ^2^ SmA [-] ^2^ CrSm CrSm 79.3 [3.1] SmA [-] ^2^ Iso Iso [-] ^2^ SmA [-] ^2^ CrSm

^1^ SmA: smectic A mesophase, Cr—Crystalline phase; CrSm—crystalline phase with smectic structure; ^2^ no transitions detected on DSC.

**Table 3 nanomaterials-12-04215-t003:** Electroactive surface area vs. electrode composition.

Electrode Type	Zn_MM:CNT:Parrafin Oil/Weight Ratio	Electroactive Surface Area/cm^2^	Electroactive/Geometrical Surface Areas Ratio
PE_01	25:1:3	0.016	0.225
PE_02	12.5:1:3	0.036	0.507
PE_03	6.25:1:3	0.051	0.720
CNT	1:3	0.117	1.53

**Table 4 nanomaterials-12-04215-t004:** Comparison of the electroanalytical performance of PE_03 with reported electrochemical sensors for UA detection.

Electrode Composition	Applied Technique	Detection Potential/V vs. SCE	Sensitivity/µA∙mM^−1^∙cm^−2^	LOD/µM	Ref.
Nafion/Uricase/Zn O micro/NWs/Au	CA	+0.80	89.74	25.6	[39]
Nitrogen-doped zinc oxide thin films	CV	+0.50	1.1	40	[40]
Nafion/ZnO QD/Uricase	CA	+0.60	4	22.97	[41]
Cationic polydiallyldimethylammonium chloride/Uricase/ZnO NPs/MWNTs/Pyrolytic graphite wafers	DPV	+0.33	393	2	[42]
ITO-r-GO-AuNPs	LSV	+0.26	0.31∙10^−3^	10.9	[43]
PE_03: 6.25 Zn_MM:1 CNT:3 Parrafinic Oil *w/w*%	MPA	−1.50	479	33	This work
		−1.20	229	25	

## Data Availability

The data presented in this study are available on request from the corresponding author.

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
