# Peer review of "Design of Nanostructured Hybrid Electrodes Based on a Liquid Crystalline Zn(II) Coordination Complex-Carbon Nanotubes Composition for the Specific Electrochemical Sensing of Uric Acid"

_nanomaterials, 2022, doi:10.3390/nano12234215_

Round 1

Reviewer 1 Report

In this paper, the complex matrix nanoplatform based on Zn(II) metallomesogen and carbon nanotubes (CNT) structures were prepared toward uric acid detection. Three kinds of weight ratio between Zn(II) metallomesogen and carbon nanotubes (CNT), PE_01, PE_02 and PE_03, were obtained by mixing the materials in different amounts. Electrochemical experiments were conducted on the as-prepared different sensors, and the largest amount of CNT PE_03 shows the best electrochemical signal for uric acid detection by cyclic voltammetry in alkaline medium. The author declares that the optimum electrochemical sensor is promising toward urine analysis. Although this manuscript has optimize the content between the Zn(II) metallomesogen and carbon nanotubes, some important issues should be clarify for electrochemical sensor.

   Some comments as follows:

  1. The morphology and structure of different weight ratio of Zn(II) metallomesogen and carbon nanotubes should be observe using SEM or TEM.

  2. Why CNT PE_03 sensor shows the superior electrochemical performance toward uric acid detection? What about the others ration? Such as PE_04, PE_05, PE_06?

3. What about the reproducibility, long-term stability and anti-interference?

4. Comparison of the electrochemical performances for as-prepared CNT PE_03 and previous reported uric acid sensor.

5. There are some mistakes about English grammar and spelling, which should carefully correct in the revision version.

6. Some related previous reported electrochemical sensors should be cited in the introduce section: Analytical Methods 7 (21), 9040-9046; Applied Surface Science 364, 434-441.

Reviewer 2 Report

In this manuscript, the authors reported the synthesis of novel Zn and CNT-based hybrids for the fabrication of electrochemical UA sensors. It is an interesting work. However, there are still some problems that should be addressed carefully. A major revision is necessary.

Special comments for the revision:

1.     In the Introduction part, it is necessary for the authors to provide more information on the fabrication of CNF-based electrodes for electrochemical sensing of analytes. More references should be cited.

2.     The authors are suggested to provide more details on the novelty and significance of this work by comparing with other previously released UA sensors.

3.     The “Materials and methods” part is very weak. The authors should use the sub-sections to introduce clearly the “Materials, synthesis of hybrids, fabrication of electrochemical electrodes, electrochemical tests, and characterization techniques”, separately.

4.     It is necessary for the authors to add a scheme to indicate clearly the preparation process of CNT-based hybrids.

5.     The morphological characterizations of Zn-MM/CNT hybrids should be carried out. Corresponding SEM and TEM images should be presented to prove successful synthesis of desired materials.

6.     The stability and selectivity of the fabricated electrochemical sensor should be tested.

7.     It is hard to evaluate the sensing performance of the fabricated sensor without the comparison with some previously similar electrochemical sensors of UA. It is necessary for the authors to add a table to compare the sensing performances.

8.     The sensing mechanism towards UA with the fabricated materials should be discussed. A scheme is suggested.

Round 2

Reviewer 1 Report

The paper can be accept in the present form.

Reviewer 2 Report

In this revised version, the authors made suitable modifications according to the comments and suggestions of all referees. Now all the questions are clear and this manuscript is recommended for publication in current form.